# Influence of smartphone addiction on sleep quality of college students: The regulatory effect of physical exercise behavior

**Weidong Zhu, Jun Liu, Hu Lou, Fanzheng Mu, Bo Li** *

College of Sports Science, Nantong University, Nantong, China

* wangqiulibo@163.com

**Data Availability Statement:** All relevant data are within the manuscript and its Supporting information files.

## Abstract

Due to the high incidence of smartphone addiction and its harmful effects on health in recent years, it has received widespread attention from society. This study aims to examine the association between smartphone addiction and sleep quality among college students, and assess the correlation with physical exercise in a non-interventional, cross-sectional study design. The study utilized data from the 2022 Chinese College Health Tracking Survey. A total of 4670 students participated in and completed the questionnaire. The test tools comprised the smartphone addiction tendency scale, the Pittsburgh Sleep Quality Index, and the physical activity rating scale. The average score of the college students' smartphone addiction was 39.230±14.931, and the proportion of college students with average and or very poor sleep quality was 52.6%. Mobile phone addiction among college students is negatively correlated with physical exercise (r = -0.101, p<0.01), and positively correlated with sleep quality (r = 0.287, p<0.01. Physical exercise had a significant regulatory effect on the behavior relationship between smartphone addiction and sleep quality ($\Delta R^2$ = 0.194, p<0.001). Smartphone addiction has a significant impact on college students' sleep quality. The higher the tendency towards smartphone addiction, the poorer the sleep quality of college students. Physical exercise plays a regulatory role in the relationship between smartphone addiction and sleep quality of college students.

## 1. Introduction

Mobile phone addiction refers to a new type of behavioral addiction in which individuals overuse smartphones to an uncontrollable extent, causing psychological, behavioral, and other problems to people [1]. According to the 51st Statistical Report on Internet Development in China, the number of Chinese netizens has reached 1.067 billion as of December 2022, of which young people aged 20–29 accounted for 14.2% [2]. With the improvement of living standards, electronic products such as smartphones, tablets, and laptops are increasingly used in the daily lives of Chinese college students. Students have spent more time in contact with smartphones due to online classes under the influence of the COVID-19 pandemic [3, 4]. At the same time, the development of technology also made electronic products such as

**Funding:** This work was supported by: Postgraduate Research & Practice Innovation Program of Jiangsu Province (KYCX24_3498), received by Weidong Zhu The General Project of Philosophy and Social Science Research in Colleges and Universities in Jiangsu Province. "Research on the promotion of physical and mental health of college students based on the improvement of physical literacy" (No: 2024SJYB1253), received by Bo Li National Social Science Fund "14th Five-Year Plan" General Project in the field of Education for the year 2021 (BLA210215), received by Hu Lou Key Project of Jiangsu Provincial Education Science Planning (B/2022/01/173), received by Bo Li.

**Competing interests:** The authors have declared that no competing interests exist.

smartphones more portable and convenient to access the internet, and the vast social world behind it often attracts college students to immerse themselves in the electronic virtual world [5].

In discussing the relationship between smartphone addiction and the sleep quality of college students, it is crucial to understand the characteristics of their stage in emerging adulthood. Emerging adulthood is a key developmental phase that involves the transition from adolescence to adulthood, during which individuals experience extended engagement in higher education, increased tolerance for premarital sex and cohabitation, and later ages of entering marriage and parenthood [6]. These changes grant them a self-focused freedom from role obligations and constraints, and satisfaction in progressing towards self-sufficiency [7]. However, the high degree of freedom and low societal role obligations at this stage may result in insufficient self-regulation in smartphone use. Excessive reliance on smartphones can not only impact their quality of life but may also lead to the development of mental health issues such as depressive symptoms and anxiety [8]. Therefore, devising effective intervention measures to reduce the negative impact of smartphone addiction is crucial for safeguarding the mental and physical health development of college students.

College students have more leisure time compared to adolescents and adults due to their special growth stage, making them a high-risk group for smartphone addiction. A study in Saudi Arabia found that the longer the use of smartphones, the deeper the dependence on it [9]. The media dependency theory also indicates that the longer the use of the media, the greater the impact it has on its users [10].

The correct use of smartphones can provide great convenience to college students' study and life, but excessive addiction to smartphone use not only reduces sleep quality but also generates negative emotions, such as fatigue and procrastination [11, 12]. Some studies have shown that the harm caused by smartphone addiction mainly includes physical problems such as headaches, blurred vision, and neck pain [13]. On the one hand, smartphone addiction can reduce the time college students spend on physical exercise, thereby decreasing their physical fitness. On the other hand, excessive use of smartphones can lead to poor sleep quality among students. A study by Orzech found that more than 72.9% of college students use smartphones for more than one hour before going to bed [14]. Pham's research also indicated that pre-sleep smartphone addiction has become an important factor affecting the sleep quality of college students [15]. Therefore, exploring the relationship between smartphone addiction and sleep quality among college students is greatly significant for their physical and mental health development.

Sleep plays an important role in the development of physical and psychological functions as part of a healthy lifestyle. Positive sleep quality is one of the necessary conditions for the development of individuals' physiological and psychological functions [16]. However, smartphone addiction is one of the most important factors that has a negative impact on sleep quality [17]. As the main force of future development of the world, college students' healthy development is related to the future of the world to some extent. Physical exercise is only considered as a measure to improve sleep quality, but few studies have explored the intrinsic relationship among smartphone addiction, sleep quality, and physical exercise. Early studies have indicated a significant correlation between smartphone addiction and a decline in sleep quality. For instance, researchers from Saudi Arabia have found that excessive smartphone use is associated with poor sleep quality [18]. A study from China has also revealed the negative impact of smartphone addiction on academic performance [19]. Additionally, there are studies indicating that smartphone addiction has a negative impact on sleep quality among the elderly population [20]. However, there is a lack of comprehensive perspective in the literature regarding how the level of physical activity might modulate this relationship.

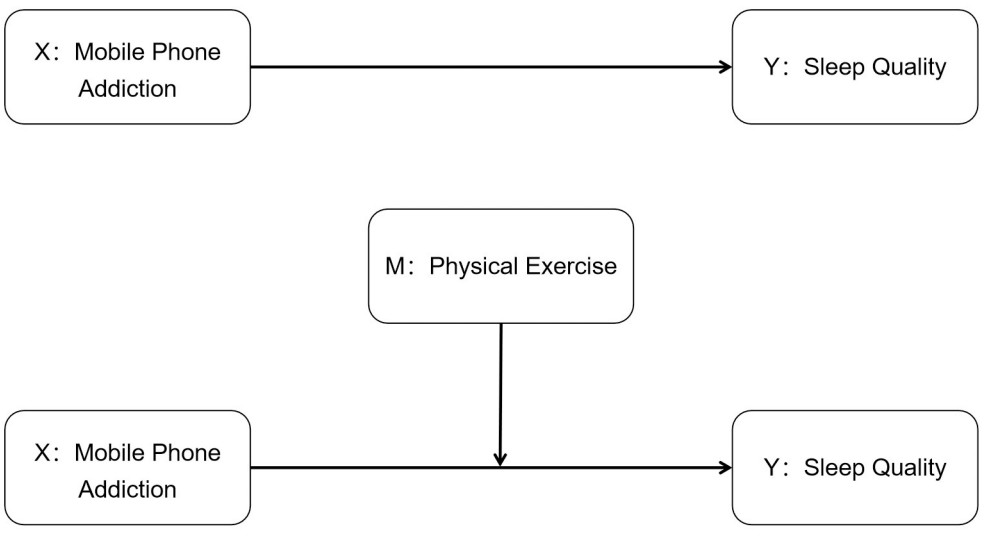

**Fig 1. The conceptual model of this study.**

In addition to the direct impacts related to smartphone addiction, physical activity is another important health behavioral factor that has been proven to positively correlate with sleep quality. Nonetheless, the interaction between smartphone addiction, physical activity levels, and sleep quality has not been fully explored in previous research. The intent of our study is to bridge this knowledge gap, with a particular focus on the unique population context in China. As one of the world's largest markets for smartphones, understanding the specific circumstances of this region is crucial for a deeper comprehension and addressing of global health concerns. This study takes college students as the research object, and taking smartphone addiction and sleep quality as entry points and introducing physical exercise as a moderating variable (as shown in Fig 1). This study aims to explore the relationship between smartphone addiction and sleep quality under the regulation of physical exercise. The primary objectives of this study are as follows:(1)To describe the patterns and extent of smartphone addiction, sleep quality, and physical exercise among Chinese college students.(2)To investigate the relationship between smartphone addiction and sleep quality in college students.(3)To examine the role of physical exercise as a potential moderating mediator in the relationship between smartphone addiction and sleep quality in college students.

## 2. Research methodology

### 2.1 Participants

This study adopted a cross-sectional research design, and eight questionnaire investigators were trained before the data collection, all of whom were undergraduate counselors from various colleges of Nantong University. Nantong University was established in 1912 and is located in Nantong City, Jiangsu Province, China. In the 2023 China University Evaluation by the China Academy of Management Sciences, it was ranked 102nd. The study utilized data from the 2022 Chinese College Health Tracking Survey, and the survey was conducted in September 2022. The participants were full-time freshmen, sophomores, juniors, and seniors of Nantong University. All students signed electronic informed consent forms before participating. All students signed an electronic informed consent form prior to participation. The participant demographics are shown in Table 1.

**Table 1. Participant demographics.**

| | Gender | | Grade | | | |
|---|---|---|---|---|---|---|
| | **Male** | **Female** | **Freshman** | **Sophomore** | **Junior** | **Senior** |
| n(4670) | 1714 | 2956 | 1606 | 1676 | 761 | 627 |
| % | 36.7 | 63.3 | 34.4 | 35.9 | 16.3 | 13.4 |

This study used a stratified random cluster sampling method, distributing 5,980 related questionnaires and receiving 4,670 valid questionnaires, with an response rate of 78%. The research plan was approved by the Ethics Committee of Nantong University (2022(70)). The minimum sample size was calculated using Formula (1) [21], where the Type I error α was set to 0.05, the allowable error δ was set to 0.01, and the sampling rate ρ was set to 0.05. Given that the total population of Yangzhou University is 48,209 people (updated data in 2023), the limited total population N was set at 75% of the student population, which is approximately 36,157 people. The calculated minimum sample size required for this study was 1,738 [22].

$$n = \frac{\left(\frac{Z_\alpha}{\delta}\right)^2 * p * (1-p)}{1 + \left[\left(\frac{Z_\alpha}{\delta}\right)^2 * p * (1-p)\right]/N} \tag{1}$$

## 2.2 Measurement methods

**2.2.1 Mobile Phone Addiction Tendency Scale (MPATS).** This study used the Mobile Phone Addiction Tendency Scale (MPATS) to measure mobile phone addiction among university students, which was compiled by Chinese scholars Xiong et al. [23]. MPATS comprises four dimensions: withdrawal symptoms, salience, social comfort, and mood changes, with a total of 16 items. Each item is scored using a five-point Likert scale: "completely not applicable," "somewhat not applicable," "neutral," "somewhat applicable," and "completely applicable," for which the scores range from 1 to 5. The highest score is 80, and the lowest is 16. A higher score indicates a greater tendency for mobile phone addiction, while a lower score indicates a lower tendency. The factor loadings of the four dimensions range from 0.51 to 0.79, with an overall variance explained of 54.3%. The confirmatory factor analysis results show the applicability of the four-factor model. The Cronbach's alpha coefficient of the total scale was 0.83, and the coefficients of the four factors ranged from 0.55 to 0.80. The test-retest reliability of the total scale was 0.91, while that of the four factors ranged from 0.75 to 0.85 [23]. The scale belongs to categorical variables.

**2.2.2 Pittsburgh Sleep Quality Index (PSQI).** The sleep quality of the university students was measured by using the Pittsburgh Sleep Quality Index (PSQI). The PSQI was compiled by Dr. Buysse, a psychiatrist at the University of Pittsburgh, along with others in 1989. The PSQI is suitable for the evaluation of sleep quality and therapeutic efficacy observation of sleep disorder patients and mental disorder patients, investigation and study of sleep quality of general population, and the evaluation tool for the study of the correlation between sleep quality and psychosomatic health. The scale is more suitable for use by the Chinese population after Liu's revisions. The scale comprises 23 items and is divided into seven components, each of which is scored on a scale of 0–3. The cumulative score of each component is the total PSQI score, which ranges from 0 to 21. The higher the total score, the worse the sleep quality becomes. The criteria for the total score are as follows: a score of 0–4 indicates good sleep quality, 5–7 indicates average sleep quality, and 8–21 indicates poor sleep quality [24]. The scale has good

reliability and validity, with a Cronbach's alpha coefficient of 0.764 [24]. The scale belongs to categorical variables.

**2.2.3 Physical Activity Rating Scale-3 (PARS-3).** The Physical Activity Rating Scale-3 (PARS-3) was compiled by Japanese scholar Hiroo Hashimoto and revised by Chinese scholar Liang [25, 26]. The scale consists of three items, and each question in the scale is divided into five levels, with scores ranging from 1 to 5. The total score is highest at 100 and lowest at 0. with 0 indicating no physical activity and 100 indicating maximum physical activity. The score criteria are as follows: low exercise level (≤19 points), moderate exercise level (20–42 points), and high exercise level (≥43 points) [25]. The PARS-3 results are a measure of sports activities and can reflect the behavioral state of college students' participation in sports at a specific time to some extent. Domestic research shows that the internal consistency reliability coefficient of the scale is 0.856, while the retest reliability coefficient is 0.820 [25]. The scale belongs to categorical variables.

## 2.3 Statistical analysis

Data processing was conducted by using IBM SPSS Statistics 26.0 and WPS Excel software (11.1.0.14309), which consisted of the following four steps: (1) Use Excel software to preprocess the data obtained from the questionnaire, and retest or delete the missing or problematic data. (2) After preprocessing the data, conduct a general analysis of the collected student data. The analysis content comprises sleep quality and physical exercise status. The Chi-square test was used to analyze the differences in physical exercise and sleep quality among students of different genders and grades. One-way ANOVA was used to analyze the differences between sleep quality and physical exercise in male and female university students of various grades based on the effect size, $\eta^2$. (3) A correlation analysis was performed to examine the correlation between the tendency of mobile phone addiction among university students (including withdrawal symptoms, highlight behavior, social comfort, and mood changes) and physical exercise, as well as sleep quality. (4) A linear regression analysis was used to test the effect of mobile phone addiction on sleep quality among university students, with physical exercise included as a moderating variable. Before calculating the moderation effect, the variables (Z-score) were standardized.

## 3. Research findings

### 3.1 General trends

Table 2 shows that the proportion of male students who engaged in moderate (21.4%) and high levels (25.3%) of physical exercise was much higher than that of female students who engaged in moderate (9.5%) and high levels (4.9%) of physical exercise. Moreover, there was a significant difference in physical exercise between male and female students ($p < 0.001$, *Cramer's V* = 0.365). From a grade perspective, a significant difference exists in physical exercise among university students in different grades ($p < 0.001$, *Cramer's V* = 0.080). The number of freshmen who had good sleep quality (52.4%) was significantly higher than that of students in other grades. No significant difference exists in sleep quality between male and female students ($p = 0.240$, *Cramer's V* = 0.400), but a significant difference exists in sleep quality among university students in different grades ($p < 0.001$, *Cramer's V* = 0.770).

In Table 3, it can be seen that there was no significant difference in mobile phone addiction between male and female students (F = 0.240, p = 0.624, $\eta^2 < 0.001$). However, a significant difference exists in mobile phone addiction among university students in different grades (F = 17.998, p<0.001, $\eta^2 < 0.001$).

**Table 2. General situation of physical exercise and sleep quality among college students.**

| | Total (n = 4670) | | Male (n = 1714) | | Female (n = 2956) | | Freshman (n = 1606) | | Sophomore (n = 1676) | | Junior (n = 761) | | Senior (n = 627) | |
|---|---|---|---|---|---|---|---|---|---|---|---|---|---|---|
| | n | % | n | % | n | % | n | % | n | % | n | % | n | % |
| Physical Exercise | | | | | | | | | | | | | | |
| Low exercise level | 3443 | 73.7 | 915 | 53.4 | 2528 | 85.5 | 1122 | 69.9 | 1284 | 76.6 | 579 | 76.1 | 458 | 73 |
| Moderate exercise level | 648 | 13.9 | 366 | 21.4 | 282 | 9.5 | 258 | 16.1 | 221 | 13.2 | 79 | 10.4 | 90 | 14.4 |
| High exercise level | 579 | 12.4 | 433 | 25.3 | 146 | 4.9 | 226 | 14.1 | 171 | 10.2 | 103 | 13.5 | 79 | 12.6 |
| $X^2$ | | | 622.538 | | | | 29.636 | | | | | | | |
| p | | | <0.001 | | | | <0.001 | | | | | | | |
| Cramer's V | | | 0.365 | | | | 0.080 | | | | | | | |
| Sleep quality | | | | | | | | | | | | | | |
| Fairly good | 2213 | 47.4 | 825 | 48.1 | 1388 | 47 | 842 | 52.4 | 726 | 43.3 | 336 | 44.2 | 309 | 49.3 |
| Average | 1745 | 37.4 | 603 | 35.2 | 1142 | 38.6 | 580 | 36.1 | 685 | 40.9 | 273 | 35.9 | 207 | 33 |
| Very poor | 712 | 15.2 | 286 | 16.7 | 426 | 14.4 | 184 | 11.5 | 265 | 15.8 | 152 | 20 | 111 | 17.7 |
| $X^2$ | | | 7.460 | | | | 55.555 | | | | | | | |
| p | | | 0.240 | | | | <0.001 | | | | | | | |
| Cramer's V | | | 0.400 | | | | 0.770 | | | | | | | |

**Table 3. Descriptive analysis of electronic addiction and physical exercise among college students.**

| | Gender | | | | | | Grade | | | | | | | | | |
|---|---|---|---|---|---|---|---|---|---|---|---|---|---|---|---|---|
| | Total (n = 4670) | | Male (n = 1714) | | Female (n = 2956) | | | | Freshman (n = 1606) | | Sophomore (n = 1676) | | Junior (n = 761) | | Senior (n = 627) | |
| | M | sd | M | sd | M | sd | P | $\eta^2$ | M | sd | M | sd | M | sd | M | sd | P | $\eta^2$ |
| Mobile addiction total score | 39.23 | 14.93 | 39.37 | 15.66 | 39.15 | 14.49 | 0.62 | <0.001 | 37.14 | 12.96 | 40.51 | 14.90 | 40.91 | 16.86 | 39.14 | 16.54 | <0.001 | 0.01 |
| Withdrawal symptoms | 15.94 | 5.82 | 15.94 | 6.01 | 15.95 | 5.71 | 0.94 | <0.001 | 15.48 | 5.32 | 16.29 | 5.75 | 16.37 | 6.41 | 15.70 | 6.38 | <0.001 | 0.01 |
| Highlight behavior | 8.69 | 4.04 | 8.93 | 4.27 | 8.54 | 3.89 | 0.00 | 0.002 | 7.78 | 3.42 | 9.18 | 4.07 | 9.34 | 4.54 | 8.90 | 4.36 | <0.001 | 0.03 |
| Social comfort | 7.71 | 3.32 | 7.63 | 3.39 | 7.75 | 3.28 | 0.26 | <0.001 | 7.45 | 3.21 | 7.89 | 3.20 | 7.92 | 3.56 | 7.62 | 3.54 | <0.001 | 0.00 |
| Mood changes | 6.90 | 3.14 | 6.87 | 3.25 | 6.91 | 3.08 | 0.72 | <0.001 | 6.43 | 2.82 | 7.15 | 3.13 | 7.29 | 3.48 | 6.93 | 3.39 | <0.001 | 0.01 |

## 3.2 Correlation analysis

Table 4 indicates a significant correlation (P<0.01) between mobile phone addiction and sleep quality among college students, with dimensions ranging from 0.255 to 0.287. Additionally, a significantly negative correlation (P<0.01) exists between sleep quality and physical exercise, with a correlation coefficient of -0.037. A significant difference (P<0.01) exists between physical exercise and mobile phone addiction, with dimensions ranging from -0.084 to -0.112. All the variables involved in Table 4 are categorical variables.

## 3.3 Test for moderating effects

Table 5 presents the results of the effect of sports exercise regulation. Model 1 shows that with sleep quality as the dependent variable, the explanation rate of independent mobile addiction and regulating sports exercise variables is 18.9%. In Model 2, the predictive ability improved by 19.4%, meaning that the contribution rate of regulating effect is 19.4% under the condition that mobile addiction independent variable and sports exercise regulation variable remain unchanged. In addition, the significant F change in both models is less than 0.001. The mobile addiction independent variable and sports exercise regulation variable have a significant impact on the sleep quality dependent variable. In the analysis of variance, the P value is less

**Table 4. Correlation analysis results.**

| | withdrawal symptoms | Highlight behavior | social comfort | mood changes | Mobile addiction total score | Physical exercise total score |
|---|---|---|---|---|---|---|
| withdrawal symptoms | | | | | | |
| Highlight behavior | 0.652** | | | | | |
| social comfort | 0.596** | 0.560** | | | | |
| mood changes | 0.649** | 0.672** | 0.569** | | | |
| Mobile addiction total score | 0.829** | 0.771** | 0.704** | 0.760** | | |
| Physical exercise total score | -0.084** | -0.092** | -0.112** | -0.095** | -0.101** | |
| PSQI total score | 0.255** | 0.284** | 0.255** | 0.283** | 0.287** | -0.037** |

** stands for $p < 0.001$.

**Table 5. Results of moderating effects.**

| | Summary table of model | | | | | ANOVA | |
|---|---|---|---|---|---|---|---|
| Model | $R^2$ | $\Delta R^2$ | df1 | df2 | Sig. F Change | F | P |
| 1 | 0.189 | 0.189 | 1 | 4668 | <0.001 | 1086.561 | <0.001 |
| 2 | 0.195 | 0.194 | 1 | 4667 | <0.001 | 564.113 | <0.001 |

**Note:** Model 1 employs sleep quality as the dependent variable in the analysis model.

Model 2 assesses the predictive ability of the combined effect of mobile addiction and the regulatory effect of sports exercise beyond Model 1.

than 0.001, indicating that the regulatory effect of sports exercise regulation variable on the mobile addiction independent variable is significant. Therefore, sports exercise is a regulatory factor for mobile addiction and sleep quality.

## 4. Discussion

Sleep quality is becoming increasingly important for people's physical health in today's fast-paced life. The research object of this paper is Chinese college students. Through the investigation and analysis of their data, we explore the internal relationship among smartphone addiction, sleep quality and physical exercise. The score of smartphone addiction among university students is 39.23±14.93. Approximately 51.9% of university students have average or poor sleep quality. Smartphone addiction among university students is negatively correlated with physical exercise and positively correlated with sleep quality. Physical exercise has a significant moderating effect on the relationship between smartphone addiction and sleep quality. Smartphone addiction among university students has a significant impact on sleep quality. The higher the tendency of smartphone addiction, the poorer the sleep quality of university students. Physical exercise plays a moderating role in the relationship between smartphone addiction and sleep quality among university students.

### 4.1 Analysis of the variability in physical exercise among college students

The proportion of male university students with moderate and high exercise levels is much higher than that of female university students. This is consistent with the findings of a previous study from the United States. Trevor Egli MS determined that the main factors driving exercise motivation among American male college students are strength, competition, and challenge, while the main factors driving exercise motivation among American female college students are weight management and appearance [27]. We believe that the reason for this phenomenon

in China may be that the exercise motivations of different genders of college students have changed. Chinese male university students engage in moderate and high exercise levels more often, to pursue strength, competition and challenge, and thus most of them choose high-intensity exercise. The feudal society in China for thousands of years has given women the stereotypical label of quiet, gentle and peaceful, which has not been completely removed. Today, the aesthetic pursuit of women in China is based on to be thin for beauty, wherein more female university students engage in low-intensity aerobic exercise to maintain their beauty of body shape, which may be the reason why there are differences in physical exercise among Chinese university students based on gender.

This study found that there were significant differences in physical exercise among university students in different grades. This is consistent with Wu's research [28], which found that senior students face with graduation and future career or study planning, and can only spend less time on physical exercise. The proportion of students in the first year engaging in moderate and high levels of exercise was slightly higher than that of students in the second, third, and fourth years. However, the proportion of freshmen was significantly lower than that of sophomore, junior, and senior students in terms of low-intensity exercise. We believe that the reasons for this phenomenon may be related to the education system in Chinese universities. students in the first year have less academic pressure and apart from the daily class time, they have no worries about other things. They have sufficient time for physical exercise without worrying about the lack of physical and mental strength which will affect other affairs. However, in the second, third, and fourth years, students not only face academic pressure, but the pressure of future career planning, further education and other aspects, which make them begin to use their extracurricular time to make up for their deficiencies. Therefore, students turn to low-intensity exercise to minimize physical exertion. In fourth year, the proportion of students engaging in moderate and high levels of exercise is higher than that of the second and third years. The reason for this may be that some fourth-year students have found jobs or obtained qualifications for further education, leading to an increase in the proportion of students engaging in moderate and high levels of exercise. This pressure is particularly prominent during the pandemic [29]. Additionally, it is crucial to understand that university students in China are in a critical evolutionary phase of transitioning from adolescence to adulthood, a period characterized by significant psychological and lifestyle changes [30]. This transition phase influences not only their academic and professional choices but also impacts their psychological well-being and daily behaviors including exercise habits. Undergraduate education in Chinese universities lasts only for four years (except for medical majors), and three out of the four years of college students enrolled in 2019 have been affected by the pandemic. Their schools cannot guarantee normal teaching, which leads to a lack of solid mastery of professional knowledge and skills, greatly increasing the students' pressure for the future. Therefore, most sophomore and junior students chose low-intensity exercise to ensure that they have sufficient time and energy to study.

## 4.2 Analysis of the differences in sleep quality among college students

This study determined that 51.9% of Chinese university students had problems with sleep quality. This is consistent with Hu's research result, which found that more than half of the sample had sleep problems [31]. According to previous studies, long-term use of mobile phones and less physical exercise may be associated with poor sleep quality among university students [32].WANG suggested that university students' sleep quality is susceptible to different risk factors, and the main determinants need to be considered when designing interventions to improve their sleep quality [33]. The university students' sleep quality can be affected by

various factors, and with the development of technology, smartphone addiction has gradually become an important factor affecting university students' sleep quality [34]. This study mainly explores the influence of smartphone addiction on university students' sleep quality, and smartphone addiction is considered as the main risk factor to investigate. Elizabeth B. Dowdell reported that a quarter of college students have a habit of using smartphones before sleep, which is associated with memory decline. Moreover, the trend of using smartphones before sleep shows an increasing tendency among college students [35]. Bartel, K study also mentioned that restricting the use of mobile phones in young people would enable them to sleep longer and have higher sleep quality [36]. All aforementioned studies have proved that smartphone addiction has an impact on sleep quality.

No significant gender differences exist in sleep quality among university students. This is different from previous studies where gender differences were found to play an important role in sleep quality, with higher sleep problem rates among females [37]. However, Zhang's research found that sleep quality among East Asian women was better than that of women in other regions of the world [38]. This phenomenon may be related to cross-regional cultural differences. In terms of grade level, the sleep quality of freshmen was significantly better than that of students in the other three grades, which is related to the education system in China. In high school, Chinese students face heavy academic pressure, and the prevalence of sleep problems among Chinese high school students was 42.3%, according to Cao's research [39]. However, in university, the decrease in academic pressure in the first year leads to an increase in students' sleep time and improvement in sleep quality. However, students begin to consider their own development and future plans starting from second year, and the increasing pressure forces them to sacrifice more time in academic and skill learning. Therefore, from the data, it can be observed that the proportion of university students with poor sleep quality decreased significantly from first to third year. The reason for the improvement in sleep quality of senior students is that some of them have already found jobs with graduation approaching, and they no longer have to worry about academic and graduation pressures. Therefore, the sleep quality of students in this phase begins to improve.

## 4.3 Analysis of the variability in mobile phone addiction among college students

No significant difference exists in mobile phone addiction between students of different genders. This is different from previous studies which indicated that females are more susceptible to mobile phone addiction [40, 41]. However, no difference exists in this paper. The possible reason for this may be that the epidemic has led to a decrease in the entertainment methods of college students. Frequent online classes, school closures and dormitory lockdowns have forced college students to spend more time indoors [42]. During this period, mobile phones became the most convenient way for communication and entertainment for students, regardless of their gender, which led to the result that no difference exists in mobile phone addiction between male and female students in this study.

A significantly positive correlation exists between the four dimensions of mobile phone addiction in college students (withdrawal symptoms, salience, social comfort, and mood changes) and their sleep quality scores. A higher score on the PSQI represents lower sleep quality, which means that the higher the degree of mobile phone addiction among college students, the worse their sleep quality would be, which is consistent with the findings of Fengd's study [43]. From December 2019 to December 2022, under the impact of COVID-19, China implemented normalized prevention and control policies, which forced many universities to adopt online teaching measures once positive cases were found in the cities where they were located.

If students were in close contact with positive cases on campus, they had to undergo a seven-day home quarantine. In quarantine, smartphones became their only means of communication and information acquisition with the outside world. Social interaction is a universal basic human motivation [44]. Saadeh and H's research investigation found that 85% of people in quarantine increased their mobile phone usage during the isolation period, with 42% using their phones for more than 6 hours per day [45]. Long-term quarantine can provide strong psychological pressure. Therefore, in long-term medical isolation situations, corresponding measures must be taken to ensure the psychological health of the quarantined individuals, avoiding negative mental problems. Previous studies have shown that sleep quality is influenced by many factors, such as lifestyle, environmental factors, work conditions, social life, economic conditions, physical health status, stress, and so on [46]. Mobile phone addiction, as an environmental factor, has a negative impact on sleep quality [47–49], which is consistent with the results of the correlation analysis in this study. Poor sleep quality can lead to depression and stress [50], affecting the health of college students in a significant negative way [51, 52]. Therefore, Chinese universities should take measures to reduce the use of smartphones by students.

## 4.4 Correlation discussion

A significantly negative correlation exists between the total score of college students' physical exercise level and their sleep quality, indicating that physical exercise may improve college students' sleep quality [53, 54]. In Dolezal's study, regardless of the type and intensity of physical activity, exercise can improve subjective sleep quality [54]. Physical exercise promotes the college students' physical and mental health. Severe mobile phone addiction is a psychological illness, and physical exercise has a significant positive impact on psychological improvement [55]. Physical exercise can reduce the impact of psychological stress on emotional states, promote emotional health [56], and provide a sense of fulfillment and satisfaction. Through physical exercise, college students can increase knowledge, develop skills, and feel a sense of fulfillment [57], which can enable them to focus their attention on real-life rather than seeking fulfillment in the virtual world.

## 4.5 Moderating effect discussion

Through the analysis of the moderating effect, physical exercise has a significant moderating effect on the impact of mobile phone addiction on college students' sleep quality. From a physiological perspective, the long-term use of smartphones by college students increases the amount of microwave radiation that the body receives. The central nervous system is the most sensitive target for microwave radiation. Therefore, microwave radiation's impact on the human body is primarily through the nervous system [58]. The main cause of neurasthenia syndrome caused by electromagnetic radiation from smartphones is long-term close contact with the body [59]. This may be the reason why sleep quality is affected by mobile phone addiction. However, an explosion of information exists in today's society, and smartphones are the most economical, convenient, and fastest way for college students to obtain external information. Continuous theory holds that the brain's mental activity is continuous, and dreams at night unconsciously reflect daytime activity [60]. It is also important to consider that university students are in a crucial developmental phase transitioning from adolescence to adulthood, characterized by significant psychological and lifestyle changes. This transition not only influences their academic and career choices but also their sleep patterns and behaviors, including smartphone use [61]. During the COVID-19 epidemic, the most common information college students obtained through short videos was related to the epidemic, increasing

their anxiety. Therefore, prolonged use of smartphones can cause college students to unconsciously recall the information they searched for about the epidemic during the day [62], which is also one of the reasons for poor sleep quality in college students. Understanding the developmental stage of university students can provide additional context to analyze how factors like smartphone addiction impact their sleep quality.

The significant moderating effect of physical exercise on sleep quality in college students may be due to the following reasons: (1) When college students engage in physical exercise, they reduce the time of using smartphones, which reduces the impact of microwave radiation on their central nervous system [58]. (2) Through physical exercise, college students gain a sense of fulfillment and satisfaction, reducing their need to seek fulfillment in the virtual world of smartphones [63]. (3) Physical exercise can release the stress that college students experience in their studies and daily life, thus improving their sleep quality [64].

The limitations of this study comprise the following: (1) Mobile phone addiction is caused by multiple factors, and there are many interfering factors in the exploration process. The impact of each variable must be explored through quasi-experimental and longitudinal studies, which can be future research directions. (2) The scope of the investigation needs to be further expanded, and the research subjects should be representative to verify the universality and rationality of this study. (3) This study only discusses the impact mechanism of physical exercise on mobile phone addiction and sleep quality, without delving into the discussion of causal relationships. (4) The study did not collect information on participant characteristics and potential confounding factors during data collection, which is a limitation of this study.

## 5. Conclusion

The proportion of university students with sleep quality problems is 51.9%, which may be related to the degree of mobile phone addiction. The proportion of male university students who engage in moderate and high levels of exercise is much higher than that of female university students, and this phenomenon may be due to differences in exercise motivation. Mobile phone addiction among university students is negatively correlated with physical exercise and positively correlated with sleep quality. The more severe the mobile phone addiction among university students, the worse their sleep quality would be. Physical exercise can significantly regulate the adverse effects of mobile phone addiction on sleep quality among university students. Although the sample size of this study is concentrated in a single school, the students admitted to this school come from different provinces and regions. Therefore, there are variations in lifestyle habits and practices, which to some extent reduces the limitations of the study. The data can provide theoretical support for improving the sleep quality of university students.

## Supporting information

**S1 Data.**
(SAV)

## Acknowledgments

We are grateful to the participants and their universities for the cooperation and participation in this study.

## Author Contributions

**Data curation:** Jun Liu, Hu Lou, Fanzheng Mu.

**Writing – original draft:** Weidong Zhu, Bo Li.

**Writing – review & editing:** Bo Li.

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
