## [Decision Letter · Decision Letter 0]

14 Aug 2023

PONE-D-23-21239Influence of Smartphone Addiction on Sleep Quality of College Students: The Regulatory Effect of Physical Exercise BehaviorPLOS ONE

Dear Dr. li,

Thank you for submitting your manuscript to PLOS ONE. After careful consideration, we feel that it has merit but does not fully meet PLOS ONE’s publication criteria as it currently stands. Therefore, we invite you to submit a revised version of the manuscript that addresses the points raised during the review process.

We look forward to receiving your revised manuscript.

Kind regards,

Md. Saiful Islam, BPH, MPH

Academic Editor

PLOS ONE

Journal Requirements:

"This study was supported by the National Social Science Fund “14th Five-Year Plan” General Project in the field of Education for the year 2021 (BLA210215)"

Reviewers' Comment

Reviewer's Responses to Questions

**Comments to the Author**

1. Is the manuscript technically sound, and do the data support the conclusions?

Reviewer #1: Yes

Reviewer #2: No

Reviewer #3: Yes

2. Has the statistical analysis been performed appropriately and rigorously? 

Reviewer #1: Yes

Reviewer #2: No

Reviewer #3: Yes

3. Have the authors made all data underlying the findings in their manuscript fully available?

Reviewer #1: Yes

Reviewer #2: Yes

Reviewer #3: Yes

4. Is the manuscript presented in an intelligible fashion and written in standard English?

Reviewer #1: Yes

Reviewer #2: Yes

Reviewer #3: No

5. Review Comments to the Author

Reviewer #1: 1.Your English writing needs further improvement. The language in submitted articles must be clear, correct, and unambiguous.

2.Your paper layout needs further beautification. For example, formula (1) is too large for the paper.

Reviewer #2: Thank you for giving me the opportunity for this review.

This study examined the effect of smartphone addiction on sleep quality in a single university in China, taking into account exercise behavior. Smartphone addiction is an emerging social issue and this study deals with an important public health topic.

While the results presented are common sense and valid, the quality of this study is low and the data does not seem to corroborate the results presented with a high degree of accuracy.

There are several major issues with this study. First, this is a cross-sectional study, and the study design does not allow for reference to causal relationships. Second, the study participants were students affiliated with a single university, making the study less representative. Third, the analysis only identified simple correlations and did not adjust for several potential confounders.

Participants

1. The authors should provide an overview of Nantong University so that readers can assess representativeness.

2. The authors should describe the study participants in detail, including in the text and a table.

3. The authors should provide an in-depth discussion of one of the major limitations of this study, representativeness, in the discussion section.

4. I did not understand the description of the sample size calculation method.

Measurement Methods

5. The authors should provide a table of data on study participant characteristics, such as obesity rate, smoking and alcohol consumption rates, and potential confounders that may influence the relationship between smartphone addiction and sleep, such as academic achievement, financial ability, commuting time to university, and living environment, if they measured these factors.

6. If the authors did not have access to the characteristics and potential confounders of the study participants, this should be noted in the discussion section as a limitation of the study.

7. The authors should cite the original article on the physical activity assessment study.

Statistical Analysis

8. The authors should adjust for confounders using multivariable analysis methods.

Reviewer #3: In this study, the authors investigate the influence of smartphone addiction on the sleep quality of college students: the regulatory effect of physical exercise behavior. The study draws attention with its sample size. My concerns about the study are listed below.

As the authors know, there are many studies conducted in this area (33. Sahin S, Ozdemir K, Unsal A, Temiz N. Evaluation of mobile phone addiction level and sleep quality in university students. Pak J Med Sci. 2013;29(4):913-8. doi: 10.12669/pjms.294.3686. PubMed PMID: WOS:000322619000005.)

More up-to-date resources in WOS could have been used (Buke, M., Egesoy, H., & Unver, F. (2021). The effect of smartphone addiction on physical activity level in sports science undergraduates. Journal of bodywork and movement therapies, 28, 530-534).

The concept of hypothesis is misused. H1-H2-H3 were questions that the authors sought answers to.

Mobile Phone Addiction Tendency Scale (MPATS) and Physical Activity 151 Rating Scale-3 (PARS-3) should be included in the summary.

Spelling and punctuation in the text should be checked. Use the dot after the closed parenthesis.

6. PLOS authors have the option to publish the peer review history of their article (what does this mean?). If published, this will include your full peer review and any attached files.

Reviewer #1: No

Reviewer #2: No

Reviewer #3: No

---

## [Author Response · Author response to Decision Letter 0]

14 Oct 2023

Dear Editors and Reviewers:

Thank you for your letter and the reviewer’s comments regarding our manuscript, “Influence of Smartphone Addiction on Sleep Quality of College Students: The Regulatory Effect of Physical Exercise Behavior”(PONE-D-23-21239).These comments have been invaluable and helpful in revising and improving our paper, as well as important guidance for our research. We have carefully studied these comments and made revisions, which we hope will be recognized. The revised parts are highlighted in red in the paper. The corrections in the paper and the responses to the reviewer’s comments are as flowing:

Reviewer #1: 

1.Your English writing needs further improvement. The language in submitted articles must be clear, correct, and unambiguous.

Response:We recognize your concern for the clarity and correctness of our English writing. We apologize for any ambiguities or errors that may have occurred. As a result, we have sought the assistance of a professional academic editing agency to ensure the accuracy and fluency of the language in this revision. We will be working closely with them to meticulously proofread and revise the paper.Thank you for your suggestion, and we will strive to ensure the language quality of the paper, as evidenced by the proof of editing provided below.

2.Your paper layout needs further beautification. For example, formula (1) is too large for the paper.

Response:We appreciate your feedback on the layout of our paper. We understand that the font size of Equation (1) is too large, making the article aesthetically unappealing. We will carefully adjust the font size and formatting to ensure that all formulas and charts are visually appealing while maintaining readability. Thank you for your suggestion, and we will strive to make improvements accordingly.

Reviewer #2：

1.This is a cross-sectional study, and the study design does not allow for reference to causal relationships.

Response:Thank you for your review and valuable feedback on our research titled "Influence of Smartphone Addiction on Sleep Quality of College Students: The Regulatory Effect of Physical Exercise Behavior" We greatly appreciate your comments and suggestions regarding the quality of research and data analysis.

We only discussed potential mechanisms of influence and did not include a discussion of causality in this research program. Additionally, we provided an explanation of the limitations of our study in the discussion section.

Thank you for your valuable feedback once again.

2. The study participants were students affiliated with a single university, making the study less representative.

Response:Thank you very much for your question about the representativeness of the sample. Although the participants in this study all come from the same university, they represent different majors and come from various provinces, cities and counties across China. Nonetheless, we recognize that the sample still has certain limitations and cannot fully represent the entire Chinese college student population.. For future studies, we will consider expanding the sample size and covering more schools and districts to improve the representativeness of the study. Thank you for your reminder and feedback.

3.The analysis only identified simple correlations and did not adjust for several potential confounders.

Response:Thank you very much for your question. This study only addresses the mechanisms of the moderating effects of physical exercise behavior and does not include explanations regarding confounding factors. Thank you for your question.

Participants

1. The authors should provide an overview of Nantong University so that readers can assess representativeness.

Response:Thank you very much for your feedback. We have included line numbers 103-106 in the article to provide additional information on the location and background of Nantong University, as well as when it was founded and its position in the education system in order to provide readers with a better understanding of the basics of Nantong University. Thank you for your suggestion.

2. The authors should describe the study participants in detail, including in the text and a table.

Response:Thank you very much for your question. We have added Table 1, which presents the sample distribution, and detailed textual explanations in lines 110-113 of the article. so that readers can have a more detailed understanding of the basic information of the study participants. We appreciate your suggestion and have made the necessary modifications.

3. The authors should provide an in-depth discussion of one of the major limitations of this study, representativeness, in the discussion section.

Response:Thank you very much for your question. We will discuss the representativeness of the sample in the discussion section. This can be found at lines 428-431 of the article.

4. I did not understand the description of the sample size calculation method.

Response:Thank you very much for your question about the sample size calculation method. The calculation method used in this article is based on the approach mentioned in the literature by Chinese scholar Shao Zhiqiang. In this method, α corresponds to the formula zα, and α is generally set to 0.05. For a two-tailed test, z0.05 is 1.96, and for a one-tailed test, z0.05 is 1.64. δ represents the allowable error, which is usually determined based on experience and should not be too large. p represents the prevalence, which is the overall population prevalence and is typically set at 0.05. The finite population size N depends on the target population, such as the total number of students in a school or patients in a hospital.

Measurement Methods

5. The authors should provide a table of data on study participant characteristics, such as obesity rate, smoking and alcohol consumption rates, and potential confounders that may influence the relationship between smartphone addiction and sleep, such as academic achievement, financial ability, commuting time to university, and living environment, if they measured these factors.

Response:Thank you very much for your question. As the questionnaire did not collect data on factors such as obesity rate, smoking and drinking rates, academic achievement and financial ability , this will be noted in the limitations of this study.

6. If the authors did not have access to the characteristics and potential confounders of the study participants, this should be noted in the discussion section as a limitation of the study.

Response:Thank you very much for your question. Since this article did not collect detailed data on features and confounding factors during data collection, we will point this out in the limitations of the discussion section,lines413-417.

7. The authors should cite the original article on the physical activity assessment study.

Response:Thank you very much for your question. We will add the original reference from Japanese scholar Hashimoto Kimio to the section on the sports activity level scale in section 2.2.3.

Statistical Analysis

8. The authors should adjust for confounders using multivariable analysis methods.

Response: Thank you very much for your question. This study only focuses on the regulatory mechanisms of physical exercise and does not involve explanations regarding confounding factors. Thank you for your question.

Reviewer #3: In this study, the authors investigate the influence of smartphone addiction on the sleep quality of college students: the regulatory effect of physical exercise behavior. The study draws attention with its sample size. My concerns about the study are listed below.

1.As the authors know, there are many studies conducted in this area (33. Sahin S, Ozdemir K, Unsal A, Temiz N. Evaluation of mobile phone addiction level and sleep quality in university students. Pak J Med Sci. 2013;29(4):913-8. doi: 10.12669/pjms.294.3686. PubMed PMID: WOS:000322619000005.)

More up-to-date resources in WOS could have been used (Buke, M., Egesoy, H., & Unver, F. (2021). The effect of smartphone addiction on physical activity level in sports science undergraduates. Journal of bodywork and movement therapies, 28, 530-534).

Response: Thank you very much for your guidance. We have replaced reference 33 in this study.

2.The concept of hypothesis is misused. H1-H2-H3 were questions that the authors sought answers to.

Response: Thank you very much for your question. This study will exclude the hypothesis section and only focus on seeking answers to the questions proposed by H1-H2-H3.

3.Mobile Phone Addiction Tendency Scale (MPATS) and Physical Activity 151 Rating Scale-3 (PARS-3) should be included in the summary.

Response: Thank you very much for your question. We will add the Mobile Addiction Tendency Scale and the Physical Activity Level Scale in the abstract, specifically at line number 31-33.

4.Spelling and punctuation in the text should be checked. Use the dot after the closed parenthesis.

Response: Thank you very much for your question. We have carefully checked the spelling and punctuation in the manuscript again. We have also sought assistance of a professional proofreading agency to ensure the coherence and correctness of the language in the article.

---

## [Decision Letter · Decision Letter 1]

6 Feb 2024

PONE-D-23-21239R1Influence of Smartphone Addiction on Sleep Quality of College Students: The Regulatory Effect of Physical Exercise BehaviorPLOS ONE

Dear Dr. li,

Thank you for submitting your manuscript to PLOS ONE. After careful consideration, we feel that it has merit but does not fully meet PLOS ONE’s publication criteria as it currently stands. Therefore, we invite you to submit a revised version of the manuscript that addresses the points raised during the review process.

I saw the comments of the reviewers, both in the previous round of review and in the present one. Some comments are somewhat conflicting, but I believe, based on their judgments, that there are minor changes that could be made to improve the work. I therefore invite you to further revise the paper, which can improve a lot if you follow these suggestions.

We look forward to receiving your revised manuscript.

Kind regards,

Giulia Ballarotto

Academic Editor

PLOS ONE

Journal Requirements:

Reviewers' comments:

Reviewer's Responses to Questions

**Comments to the Author**

1. If the authors have adequately addressed your comments raised in a previous round of review and you feel that this manuscript is now acceptable for publication, you may indicate that here to bypass the “Comments to the Author” section, enter your conflict of interest statement in the “Confidential to Editor” section, and submit your "Accept" recommendation.

Reviewer #3: All comments have been addressed

Reviewer #4: (No Response)

2. Is the manuscript technically sound, and do the data support the conclusions?

Reviewer #3: Partly

Reviewer #4: Yes

3. Has the statistical analysis been performed appropriately and rigorously? 

Reviewer #3: Yes

Reviewer #4: Yes

4. Have the authors made all data underlying the findings in their manuscript fully available?

Reviewer #3: Yes

Reviewer #4: No

5. Is the manuscript presented in an intelligible fashion and written in standard English?

Reviewer #3: Yes

Reviewer #4: Yes

6. Review Comments to the Author

Reviewer #3: In this study, the authors investigate the influence of smartphone addiction on the sleep quality of college students: the regulatory effect of physical exercise behavior.

The authors have completed the necessary corrections.

Kind regards

Reviewer #4: Abstract:

"explore its regulating effect by introducing physical exercise" This refer as if the study an interventional one. Please rephrase in a way that make it a cross sectional study.

"r=-0.101", here do you mean r-squared of the regression? if not, please report r-squared instead of r value.

Introduction:

-"problems to people.[1]" please adjust the reference style.

- "generates negative emotions, such as fatigue and procrastination." where is the reference?

- "The current study on smartphone addiction mainly focus on mental health and

hygiene", Do you mean your study? Do you mean mental hygiene? Clarify please.

- Please include studies that previously assessed the relationship between smartphone addiction and sleep quality (such as https://www.ncbi.nlm.nih.gov/pmc/articles/PMC9879389/), and those also assessed physical activity and its effect on sleep quality? Is there any study in China? Please highlight in the introduction to clarify the gap in knowledge.

- "What are the characteristics of contemporary college students' smartphone addiction, sleep quality, and physical exercise?" This is not the correct way to write a scientific hypothesis. In fact, the all the of them need to be rewritten or to change it into study objective.

-remove the word contemporary and rephrase the sentence.

Methods:

- "The scales used in the" do you mean questionnaire? replace.

- "effective rate of 78%." change it into response rate.

- "Given that the total population is 48,209 people" the total population of all university students in China, or in that university or what? if you have the reference that will be great.

Research Findings

- please include a table about demographic data of the participants.

- Also, I think you have to put a descriptive statistics to all the variables, in continuous and categorical format before going into analysis. This will answer the research hypothesis number 1 in your research.

- Table 1 design is little confusing, I wish you can adjust it so it can be read easily. For example, you have in the row three sections total, gender and academic year, so please make that clear.

- Table 2 is so confusing with the numbers, please either change the layout, or use 1 or 2 decimals maximum in the table. Remove F-value, as it is confusing.

- In Table 4 please put footnote for what model 1 include, and what model 2 include.

- In Table 4 please clarify if you used the data in its continuous format or categorical one, because that was not clear to reader.

Discussion:

- "previous studies[16]." in what country?

-"Trevor Egli MS" do you mean that this is reference 16 please put the reference here.

- In discussion from line 248-258, they were written without reference, if this is the author opinion, I think you should write, it is suggested or similar terms. This is applied to many parts in the discussion, where many sentences without reference. So I suggest that write what make it clear that this is your suggestion

- Your discussion is long, and I think you need to make subheading for it.

7. PLOS authors have the option to publish the peer review history of their article (what does this mean?). If published, this will include your full peer review and any attached files.

Reviewer #3: No

Reviewer #4: No

---

## [Author Response · Author response to Decision Letter 1]

12 Feb 2024

Dear Editors and Reviewers:

Thank you for your letter and the reviewer’s comments regarding our manuscript, “Influence of Smartphone Addiction on Sleep Quality of College Students: The Regulatory Effect of Physical Exercise Behavior”(PONE-D-23-21239).These comments have been invaluable and helpful in revising and improving our paper, as well as important guidance for our research. We have carefully studied these comments and made revisions, which we hope will be recognized. The revised parts are highlighted in red in the paper. The corrections in the paper and the responses to the reviewer’s comments are as flowing:

Reviewer #4: Abstract:

"explore its regulating effect by introducing physical exercise" This refer as if the study an interventional one. Please rephrase in a way that make it a cross sectional study.

Response: Thank you for your insightful comments and for emphasizing the need for clarity regarding our study design. As you have correctly pointed out, the original wording suggested an interventional approach, which was not in line with the intent of our study.

I have revised the language in the manuscript to more accurately convey that our research is a cross-sectional analysis. We have rephrased the purpose of our study to underscore the examination of associations rather than implying any intervention. Moreover, I have highlighted the correlation between physical exercise and the relationship between smartphone addiction and sleep quality to further clarify that our study design is observational in nature.

I hope these modifications address your concerns and faithfully communicate the cross-sectional nature of our study. Please feel free to make any further suggestions if necessary. For these changes, please refer to lines 26-30.

"r=-0.101", here do you mean r-squared of the regression? if not, please report r-squared instead of r value.

Response: Thank you for your inquiry. The "r=-0.101" mentioned here refers to the correlation coefficient result, not the r-squared value of the regression. You can find the report on r-squared in lines 37 to 39 of the revised manuscript. Should you have any further questions or require additional information, please feel free to let me know.

Introduction:

-"problems to people.[1]" please adjust the reference style.

Thank you for bringing this to my attention. I have corrected the reference style as per your suggestion. The updated reference can now be found in the revised manuscript at line number 49.

- "generates negative emotions, such as fatigue and procrastination." where is the reference?

Response:Thank you for your question and for pointing out the oversight. We have added the appropriate reference for this statement. Please refer to line 67 of the revised manuscript for the specific correction and the added citation. If there are any other concerns or clarifications needed, do not hesitate to contact us.

- "The current study on smartphone addiction mainly focus on mental health and

hygiene", Do you mean your study? Do you mean mental hygiene? Clarify please.

Response:Thank you for seeking clarification. When I referred to "The current study on smartphone addiction mainly focus on mental health and hygiene," I was intending to summarize the general trend of research in the area of smartphone addiction. Upon further examination, we recognized that there is insufficient evidence in the literature to substantiate the focus on mental health and hygiene specifically in relation to smartphone addiction. Consequently, we have opted to eliminate this statement from our manuscript to ensure accuracy. This change can be found in the revised version, where lines 67-69 have been removed.

- Please include studies that previously assessed the relationship between smartphone addiction and sleep quality (such as https://www.ncbi.nlm.nih.gov/pmc/articles/PMC9879389/), and those also assessed physical activity and its effect on sleep quality? Is there any study in China? Please highlight in the introduction to clarify the gap in knowledge.

Response: Thank you for your insightful comments and your suggestion to include prior studies that assessed the relationship between smartphone addiction and sleep quality, as well as the intersection of physical activity and its effect on sleep quality.

In response to your request, I will enrich the introduction section by discussing several pivotal studies that have investigated these important themes. For example, I will reference the study you provided (https://www.ncbi.nlm.nih.gov/pmc/articles/PMC9879389/), which offers a comprehensive analysis of how excessive smartphone use may detrimentally affect sleep quality.

Furthermore, I will make an effort to highlight research conducted in China regarding this topic. Indeed, scholars in China have undertaken numerous studies examining the effects of smartphone addiction on various aspects of health and daily life, including sleep quality. I will identify and include relevant Chinese studies to underline the existing research in the area, which will help to illustrate the current gap in knowledge that our study aims to address.

By integrating these studies, the introduction will clearly state the contribution of our research within the broader context of existing literature. This will not only fortify the backdrop against which our study stands but will also elucidate the necessity for further research to bridge the knowledge gap identified.For the relevant revisions, please refer to lines 86-92. 

- "What are the characteristics of contemporary college students' smartphone addiction, sleep quality, and physical exercise?" This is not the correct way to write a scientific hypothesis. In fact, the all the of them need to be rewritten or to change it into study objective.

Response: Thank you for your keen observation regarding the construction of our research questions and hypotheses. Upon reflection, we concede that the initial formulation of our hypotheses more closely resembled questions, which could lead to misinterpretation and are not in line with the traditional scientific format.

In light of your comment, we have revised our study objectives to clearly delineate our aims and dispel any confusion. The hypotheses have been reformulated into objective statements that clearly chart the course of our investigation and lay the groundwork for operationalization and empirical testing. We believe these adjustments will elucidate the intent and scope of our research, ensuring that our objectives are more precise and methodologically robust.

We are grateful for the opportunity to integrate your invaluable feedback, which has undoubtedly enriched the framework of our research. We trust that these revisions will meet with your approval and will elevate the quality of our study. The pertinent revisions can be referenced on lines 107-112.

-remove the word contemporary and rephrase the sentence.

Response: Thank you for your suggestion. We have removed the word "contemporary" and rephrased the sentence to better reflect our intended meaning. The revised sentence maintains the essence of the original statement while ensuring clarity and conciseness. We appreciate your attention to detail and agree that this modification improves the manuscript. Please see the revised sentence on line 109 of the updated document.

Methods:

- "The scales used in the" do you mean questionnaire? replace.

Response: Thank you for your comment. Indeed, when we referred to "the used scales," we were specifically alluding to the scales within the questionnaire. We have made the necessary correction in the text to express this unambiguously. The term "scales" previously mentioned in the text has now been replaced with "questionnaire." Please refer to line 122 in the revised manuscript for this correction. We appreciate your meticulous review and valuable feedback.

- "effective rate of 78%." change it into response rate.

Response: Thank you for your attentive reading of our manuscript and the suggestion to clarify our terminology. We agree that the term “effective rate of 78%” could be misunderstood and that “response rate” is more appropriate and commonly utilized in this context.

We have updated the manuscript to reflect this change. The phrase “effective rate of 78%” has now been revised to “response rate of 78%” to ensure clarity and align with standard research terminology.

Please refer to the updated line 131 in the revised manuscript where this correction has been applied.

We appreciate your valuable input and believe that this modification improves the accuracy and readability of our study.

- "Given that the total population is 48,209 people" the total population of all university students in China, or in that university or what? if you have the reference that will be great.

Response: Thank you very much for your question. The total population of 48,209 people mentioned here refers to the total number of enrolled students at Yangzhou University as of 2023, not the total number of university students in China. You can find the relevant information on the total number of students on the official website of Yangzhou University, which is http://www.yzu.edu.cn/xxgk/xxjj.htm. We apologize for not making this clear in our previous description in the article, and we have made the necessary revisions in the revised manuscript. Please refer to the updated line number 135.

Research Findings

- please include a table about demographic data of the participants.

Response: Thank you for your inquiry. I can confirm that the demographic data of the participants is indeed included in the article. We have organized this information into three sections: gender, academic level, and age. Please refer to line 128 for the relevant revisions.

- Also, I think you have to put a descriptive statistics to all the variables, in continuous and categorical format before going into analysis. This will answer the research hypothesis number 1 in your research.

Response: Thank you very much for your query. We have indeed classified the variables in the scale into their respective categories. For more detailed information, please refer to the revised manuscript at lines 157-158, 173-174, and 186. We believe this classification aligns with the requirements to adequately address research hypothesis number 1.

- Table 1 design is little confusing, I wish you can adjust it so it can be read easily. For example, you have in the row three sections total, gender and academic year, so please make that clear.

Response: Thank you for your meticulous review and valuable suggestions. We have made a series of improvements to Table 1 as per your guidance. Firstly, we have enhanced the distinction among the various categorization fields, ensuring that the “Total,” “Gender,” and “Grade Level” sections are clear and well-differentiated. Additionally, to further improve the table’s conciseness and readability, we have removed the “Age Group” information.

Please review the revised manuscript at your convenience. We highly value your feedback and hope that our modifications meet your approval.

We are sincerely grateful for your contribution to our work and look forward to your response.

- Table 2 is so confusing with the numbers, please either change the layout, or use 1 or 2 decimals maximum in the table. Remove F-value, as it is confusing.

Response: Thank you very much for your feedback. In reviewing Table 2, we noticed that it did not list the F-values; hence, we presume you might be referring to the issue with the F-values in Table 3. Following your suggestion, we have removed the F-values from Table 3 and adjusted the numeric values to two decimal places for increased clarity and conciseness.

We hope these changes address your concerns and enhance the readability of the data presentation. For details on the revisions, please refer to line number 225.

Thank you again for your valuable comments.

 - In Table 4 please put footnote for what model 1 include, and what model 2 include.

Response: Thank you for inquiring about the contents of Table 4. As per your request, we will ensure the provision of appropriate footnotes detailing the components of Model 1 and Model 2 to enhance the clarity of our analysis for the reader's comprehension. We will make certain that these details are succinctly summarized in the footnotes to aid readers in better understanding the composition and outcomes of the models in our study. For specific modifications, please refer to lines 253-255.

- In Table 4 please clarify if you used the data in its continuous format or categorical one, because that was not clear to reader.

Response: Thank you very much for your inquiry regarding the nature of the data used in Table 4. To clarify, the data presented in Table 4 have been processed as categorical variables. We are extremely grateful for your attention to detail and understand the importance of clarifying this to our readers. To eliminate any ambiguity, we will revise the manuscript to explicitly state that the variables in Table 4 have been analyzed as categorical variables. We will make the necessary changes at lines 234-235 of the manuscript.

Discussion:

- "previous studies[16]." in what country?

Response: Thank you for your question regarding “previous studies[16].” I realize the importance of providing the geographic context of referenced studies for the reader’s understanding.

In response to your query, the previous studies mentioned are based in the United States. To clarify this in our manuscript, we have now included the country where the referenced research was conducted. You can find this update in lines 273-274 of the manuscript.

I appreciate your keen eye for detail and the opportunity to make our manuscript as informative and precise as possible.

-"Trevor Egli MS" do you mean that this is reference 16 please put the reference here.

Response: Thank you for your question, and I apologize for the oversight. I have now corrected the placement of the reference. Please see line 278 in the manuscript for the specific citation.

- In discussion from line 248-258, they were written without reference, if this is the author opinion, I think you should write, it is suggested or similar terms. This is applied to many parts in the discussion, where many sentences without reference. So I suggest that write what make it clear that this is your suggestion

Response: Thank you for your feedback. I understand your concerns regarding the lack of references for certain opinions expressed between lines 248-258 of the discussion section. I will take your suggestion and make it clear that these are our suggestions or personal viewpoints. I will address this issue throughout the relevant parts of the discussion to avoid confusion for the readers. For sentences without references, I will add phrases such as "we suggest" or "based on our observation" to clearly indicate that it reflects the authors' perspectives. Once again, I appreciate your valuable comments. For specific modifications, please refer to the manuscript lines 278-279 and 296-297.

- Your discussion is long, and I think you need to make subheading for it.

Response:Thank you for your constructive feedback on the discussion section of our manuscript. I appreciate your suggestion to incorporate subheadings to improve the readability and structure of our text.

I agree that the discussion section is rich in content, and adding subheadings will help readers better navigate and understand the various arguments and conclusions we have presented. We will revise this part by adding relevant subheadings that reflect the core points and themes of our discussion. Such restructuring will undoubtedly make the content more understandable and focused.

We will ensure that each subheading is short and descriptive, allowing readers to quickly grasp the main idea of the subsequent paragraphs. We believe these changes will enhance the overall clarity of the manuscript, and we thank you for pointing this out.

For specific modifications, please refer to lines 272-273, 319-320, 360-361, 398, and 411 of the manuscript.

---

## [Decision Letter · Decision Letter 2]

9 May 2024

PONE-D-23-21239R2Influence of Smartphone Addiction on Sleep Quality of College Students: The Regulatory Effect of Physical Exercise BehaviorPLOS ONE

Dear Dr. Li,

Thank you for submitting your manuscript to PLOS ONE. After careful consideration, we feel that it has merit but does not fully meet PLOS ONE’s publication criteria as it currently stands. Therefore, we invite you to submit a revised version of the manuscript that addresses the points raised during the review process.

We look forward to receiving your revised manuscript.

Kind regards,

Sohel Ahmed, BPT, MPT, MDMR

Academic Editor

PLOS ONE

Journal Requirements:

Reviewers' comments:

Reviewer's Responses to Questions

**Comments to the Author**

1. If the authors have adequately addressed your comments raised in a previous round of review and you feel that this manuscript is now acceptable for publication, you may indicate that here to bypass the “Comments to the Author” section, enter your conflict of interest statement in the “Confidential to Editor” section, and submit your "Accept" recommendation.

Reviewer #3: All comments have been addressed

Reviewer #5: All comments have been addressed

2. Is the manuscript technically sound, and do the data support the conclusions?

Reviewer #3: Yes

Reviewer #5: Yes

3. Has the statistical analysis been performed appropriately and rigorously? 

Reviewer #3: Yes

Reviewer #5: Yes

4. Have the authors made all data underlying the findings in their manuscript fully available?

Reviewer #3: Yes

Reviewer #5: Yes

5. Is the manuscript presented in an intelligible fashion and written in standard English?

Reviewer #3: Yes

Reviewer #5: Yes

6. Review Comments to the Author

Reviewer #3: This study examined the effect of smartphone addiction on college students' sleep quality: the moderating effect of physical exercise behavior. The corrections I requested have been made, thank you.

Reviewer #5: Thank you very much for the possibility to review the study "Influence of Smartphone Addiction on Sleep Quality of College Students: The Regulatory Effect of Physical Exercise Behavior". The work is interesting and the revisions carried out have allowed an improvement of the work. I think it has improved significantly. At the same time, however, there are some aspects that should be improved. In particular, the authors focused on university students, highlighting the association between smartphone addiction and the quality of sleep. It would be useful to be able to discuss the results of the study (but also an in-depth analysis in the introductory part) of the evolutionary phase in which the participants of the study find themselves. In particular, there is a large literature on emerging adulthood that should be explored in order to better understand the results of the study. Some specific studies are recommended:

- Arnett, J. J. (2007). Emerging adulthood: What is it, and what is it good for?. Child development perspectives, 1(2), 68-73.

- Ballarotto, G., Marzilli, E., Cerniglia, L., Cimino, S., & Tambelli, R. (2021). How does psychological distress due to the COVID-19 pandemic impact on internet addiction and Instagram addiction in emerging adults?. International journal of environmental research and public health, 18(21), 11382.

- Hochberg, Z. E., & Konner, M. (2020). Emerging adulthood, a pre-adult life-history stage. Frontiers in endocrinology, 10, 918.

- Tambelli, R., Cimino, S., Marzilli, E., Ballarotto, G., & Cerniglia, L. (2021). Late adolescents’ attachment to parents and peers and psychological distress resulting from COVID-19. A study on the mediation role of alexithymia. International Journal of Environmental Research and Public Health, 18(20), 10649.

- Griffioen, N., Scholten, H., Lichtwarck-Aschoff, A., Van Rooij, M., & Granic, I. (2021). Everyone does it—differently: A window into emerging adults’ smartphone use. Humanities and Social Sciences Communications, 8(1), 1-11.

7. PLOS authors have the option to publish the peer review history of their article (what does this mean?). If published, this will include your full peer review and any attached files.

Reviewer #3: No

Reviewer #5: No

---

## [Author Response · Author response to Decision Letter 2]

16 May 2024

Reviewer #3: This study examined the effect of smartphone addiction on college students' sleep quality: the moderating effect of physical exercise behavior. The corrections I requested have been made, thank you.

Response: Thank you for your feedback. I am pleased to confirm that the necessary corrections have been made. Should you have any further queries or require additional clarifications, please feel free to contact us.

Reviewer #5: Thank you very much for the possibility to review the study "Influence of Smartphone Addiction on Sleep Quality of College Students: The Regulatory Effect of Physical Exercise Behavior". The work is interesting and the revisions carried out have allowed an improvement of the work. I think it has improved significantly. At the same time, however, there are some aspects that should be improved. In particular, the authors focused on university students, highlighting the association between smartphone addiction and the quality of sleep. It would be useful to be able to discuss the results of the study (but also an in-depth analysis in the introductory part) of the evolutionary phase in which the participants of the study find themselves. In particular, there is a large literature on emerging adulthood that should be explored in order to better understand the results of the study. Some specific studies are recommended:

- Arnett, J. J. (2007). Emerging adulthood: What is it, and what is it good for?. Child development perspectives, 1(2), 68-73.

- Ballarotto, G., Marzilli, E., Cerniglia, L., Cimino, S., & Tambelli, R. (2021). How does psychological distress due to the COVID-19 pandemic impact on internet addiction and Instagram addiction in emerging adults?. International journal of environmental research and public health, 18(21), 11382.

- Hochberg, Z. E., & Konner, M. (2020). Emerging adulthood, a pre-adult life-history stage. Frontiers in endocrinology, 10, 918.

- Tambelli, R., Cimino, S., Marzilli, E., Ballarotto, G., & Cerniglia, L. (2021). Late adolescents’ attachment to parents and peers and psychological distress resulting from COVID-19. A study on the mediation role of alexithymia. International Journal of Environmental Research and Public Health, 18(20), 10649.

- Griffioen, N., Scholten, H., Lichtwarck-Aschoff, A., Van Rooij, M., & Granic, I. (2021). Everyone does it—differently: A window into emerging adults’ smartphone use. Humanities and Social Sciences Communications, 8(1), 1-11.

Response: we thank you for your detailed review and valuable suggestions on our research. Based on your feedback, we have conducted an in-depth analysis of the introduction section and integrated the literature you recommended to enhance our understanding and background of the study topic. These revisions not only deepened the theoretical depth of our paper but also provided a more solid theoretical foundation for our research findings. Once again, thank you for your careful guidance and suggestions, which have been critical in enhancing the quality and depth of our study. Please refer to lines 59-73, 325-330, 442-447, and 452-453 of the article for specific modifications.

---

## [Decision Letter · Decision Letter 3]

2 Jul 2024

Influence of Smartphone Addiction on Sleep Quality of College Students: The Regulatory Effect of Physical Exercise Behavior

PONE-D-23-21239R3

Dear Dr. Bo li

We’re pleased to inform you that your manuscript has been judged scientifically suitable for publication and will be formally accepted for publication once it meets all outstanding technical requirements.

Kind regards,

Sohel Ahmed, BPT, MPT, MDMR

Academic Editor

PLOS ONE

Additional Editor Comments (optional):

Reviewers' comments:

Reviewer's Responses to Questions

**Comments to the Author**

1. If the authors have adequately addressed your comments raised in a previous round of review and you feel that this manuscript is now acceptable for publication, you may indicate that here to bypass the “Comments to the Author” section, enter your conflict of interest statement in the “Confidential to Editor” section, and submit your "Accept" recommendation.

Reviewer #3: All comments have been addressed

Reviewer #6: All comments have been addressed

2. Is the manuscript technically sound, and do the data support the conclusions?

Reviewer #3: Yes

Reviewer #6: Yes

3. Has the statistical analysis been performed appropriately and rigorously? 

Reviewer #3: Yes

Reviewer #6: Yes

4. Have the authors made all data underlying the findings in their manuscript fully available?

Reviewer #3: Yes

Reviewer #6: Yes

5. Is the manuscript presented in an intelligible fashion and written in standard English?

Reviewer #3: Yes

Reviewer #6: Yes

6. Review Comments to the Author

Reviewer #3: In this study, they examined the effect of smartphone addiction on the sleep quality of university students. Thanks for the corrections. .

Reviewer #6: Thank you, authors, for revising the manuscript in response to reviewer comments and queries. The manuscript has been improved satisfactorily.

The English language has been revised to make it more understandable. However, there is scope for further editing.

7. PLOS authors have the option to publish the peer review history of their article (what does this mean?). If published, this will include your full peer review and any attached files.

Reviewer #3: No

Reviewer #6: No

---

## [Editor Report · Acceptance letter]

17 Jul 2024

PONE-D-23-21239R3 

PLOS ONE

Dear Dr. Li, 

I'm pleased to inform you that your manuscript has been deemed suitable for publication in PLOS ONE. Congratulations! Your manuscript is now being handed over to our production team.

Kind regards, 

on behalf of

Dr. Sohel Ahmed 

Academic Editor

PLOS ONE